# Fluoroquinolone Resistance in *Escherichia coli* Causing Community-Acquired Urinary Tract Infections: A Systematic Review

**DOI:** 10.3390/microorganisms12112320

**Published:** 2024-11-15

**Authors:** Ana P. Ruiz-Lievano, Fernando Cervantes-Flores, Alessandro Nava-Torres, Paulo J. Carbajal-Morales, Luisa F. Villaseñor-Garcia, Maria G. Zavala-Cerna

**Affiliations:** 1Facultad de Medicina Humana, Universidad Autónoma de Chiapas, Tuxtla Gutiérrez 29050, Chiapas, Mexico; ana.ruiz02@unach.mx; 2Facultad de Medicina, Benemérita Universidad Autónoma de Puebla, Puebla 72410, Puebla, Mexico; fernando.cervantesf@alumno.buap.mx; 3Unidad Académica Ciencias de la Salud, Universidad Autónoma de Guadalajara, Zapopan 45129, Jalisco, Mexico; alessandro.nava@edu.uag.mx (A.N.-T.); paulo.carbajal@edu.uag.mx (P.J.C.-M.); luisa.villasenor@edu.uag.mx (L.F.V.-G.)

**Keywords:** fluoroquinolone resistance, urinary tract infections, systematic review, *Escherichia coli*, community-acquired bacterial infection

## Abstract

Community-acquired urinary tract infections account for 15% of all outpatient use of antibiotics, and women are primarily affected; the major causative microorganism is uropathogenic *Escherichia coli* (*E. coli*). Treatment is indicated for cystitis and pyelonephritis and includes B-lactams (amoxicillin-clavulanic acid or third-generation cephalosporins), fluoroquinolones (ciprofloxacin or levofloxacin), nitrofurantoin, fosfomycin, and trimethoprim–sulfamethoxazole. Resistance to antibiotic treatment is of concern; several mechanisms have been associated with the acquisition of genes that confer antimicrobial resistance to fluoroquinolones, which are often associated with other patterns of resistance, especially in extended-spectrum beta-lactamase (ESBL) producers. Several studies have addressed the prevalence of uropathogens producing ESBLs, but only a few have focused on fluoroquinolone resistance, and, to our knowledge, none have addressed the prevalence of phylotypes or genes responsible for antimicrobial resistance to fluoroquinolones. The focus of the present review was to analyze recently published papers that described the *E. coli* phylotype causing community-acquired UTIs in association with fluoroquinolone resistance.

## 1. Introduction

Urinary tract infections (UTIs) account for approximately 3 million visits to health services annually and are responsible for up to 15% of antibiotic use in outpatients in the United States [1]. UTIs are considered the most prevalent community-acquired (CA) bacterial infection [2]. In terms of frequency, women are affected primarily, and infections are majorly caused by *Escherichia coli* (*E. coli*), *Enterococcus*, *Staphylococcus*, *Proteus*, *Klebsiella*, and *Pseudomonas*, with uropathogenic *E. coli* (UPEC) being the causative agent in more than three-quarters of the reported UTIs annually [3,4]. A CA UTI is defined as one presenting in outpatient clinics, primary care, or the emergency department, without prior hospitalization for the index episode, excluding patients in residential/long-term-care facilities. UTIs are classified as either complicated or uncomplicated: an uncomplicated UTI (uUTI) is an episode of cystitis in a woman who is not pregnant, is not immunocompromised, has no anatomical or functional abnormalities of the urogenital tract, and does not exhibit signs of tissue invasion or systemic effects. The remaining UTIs are complicated; this distinction has implications for therapy due to complications and treatment failure risk in complicated UTIs [5]. As a rule, a man with a UTI is considered complicated due to frequent involvement of the prostate [6]; however, young males without systemic symptoms or those with a medical history with no suggestion of a causative factor might be considered to have uUTIs. Nevertheless, this presentation is uncommon [7].

Treatment is indicated for cystitis and pyelonephritis and includes B-lactams (amoxicillin-clavulanic acid or third-generation cephalosporins), fluoroquinolones (ciprofloxacin or levofloxacin), nitrofurantoin, fosfomycin, and trimethoprim–sulfamethoxazole [8]. UTIs are the main cause of antibiotic consumption, which is a concern since the widespread use of antibiotics has led to increased antimicrobial resistance in UPEC and other uropathogens, which has led to complications in the treatment, prevention, and management of recurrent UTIs. Fluoroquinolones have the advantages of availability in both oral and parenteral forms, good absorption, and high urinary excretion rates [9] (Figure 1). However, owing to resistance, the available guidelines of the Infectious Diseases Society of America (IDSA), published in 2010, recommend the use of nitrofurantoin, fosfomycin, and trimethoprim–sulfamethoxazole for empirical therapy of acute uncomplicated cystitis [10]. The European Association of Urology (EAU) guidelines recommend not using aminopenicillins or fluoroquinolones to treat uncomplicated cystitis and recommend the use of fosfomycin and nitrofurantoin as first-line treatments. The use of fluoroquinolones is restricted to men in accordance with local susceptibility testing [11]. Furthermore, the European Centre for Disease Prevention (ECDC) has identified the misuse of antibiotics as one of the leading causes of antimicrobial resistance. According to the ECDC, misuse occurs in three main scenarios: (i) unnecessary prescription of antibiotics for viral infections, (ii) prescription of broad-spectrum antibiotics because of ignorance of the causative pathogen, and (iii) incorrect doses, frequencies, or treatment durations of antibiotic administration [12].

In 2016 and 2018, respectively, the American FDA and European Medicines Agency (EMA) warned against the use of quinolones in non-serious infections, including UTIs, due to the presence of disabling and irreversible effects associated with their use, such as tendinitis, tendon rupture, peripheral neuropathy, and joint disorders [13,14].

Fluoroquinolone resistance can occur as a result of either of the following three mechanisms: (1) chromosomal mutations that change the targets of the drug, reducing its efficacy, such as the genes that encode *gyrA* (a subunit from DNA gyrase) or *parC* (subunit from topoisomerase IV); (2) mutations related to reducing the drug concentration in the bacterial cytoplasm through the overexpression of efflux pumps and the downregulation of porins; and (3) the expression of genes that encode plasmid-mediated quinolone resistance proteins (PMQRs), such as *qnr*, *aac(6′)-lb-cr*, *quepA*, and *oqxAB* [15] (Figure 2). Frequently, resistance is caused by more than one of these mechanisms.

PMQR mechanisms usually confer low-level resistance, but either alone or with chromosomal target mutations, they widen the selection window, increase the minimum inhibitory concentration (MIC) of the strains, and facilitate the selection of higher levels of resistance. Furthermore, plasmids harboring genes encoding proteins related to resistance to other antimicrobials, such as beta-lactams and aminoglycosides, are present [16].

Antimicrobial resistance genes (ARGs) can be transferred to *E. coli* strains via mobile genetic elements, which include transposons, plasmids, sequence insertion, and genomic islands [17]. Regardless of the mechanism, the transmission of these elements leads to recombination among different *E. coli* strains. The current classification for identifying the phylogenetic order of *E. coli* strains is based on the presence or absence of four genetic sequences, namely, *chuA*, *yjaA*, *TspE4*, and *arpA*, into phylotypes A, B1, B2, D, C, E, F, G, and *Escherichia* cryptic clade I [18]. Strains of the UPEC pathotype usually belong to phylotypes B2 and D, but intestinal pathogenic and commensal *E. coli* strains belong to A and B1 [19].

Another factor related to the development of resistance in *E. coli* is the use of antibiotics in veterinary medicine for food-producing animals. As a member of the gut microbiota, *E. coli* that is resistant to antibiotics can be transmitted through the food chain or directly to humans. Alternatively, the animal product itself could contain antibiotic residues that promote contact with humans and the development of resistant bacteria [20]. This last factor has been more frequently associated with the production of extended-spectrum beta-lactamases (ESBLs).

Several reviews have addressed the prevalence of antimicrobial resistance to beta-lactams in association with the production of ESBLs, but only a few have focused on fluoroquinolone resistance, and, to our knowledge, none have addressed the prevalence of isolate mutations that encode for proteins related to fluoroquinolone resistance. The focus of the present review was to analyze recently published papers that described the *E. coli* phylotype causing community-acquired UTIs in association with fluoroquinolone resistance.

## 2. Materials and Methods

This systematic review was developed according to the PRISMA guidelines [21]. We used PubMed to conduct searches and identify candidate studies to be included in the review. Keywords were obtained from MESH terminology, and we used a combination of several terms to conduct searches. The terms included “community acquired infection” AND “urinary tract infection” AND “fluoroquinolone” AND “antimicrobial drug resistance” AND “antimicrobial resistance genes”. Papers were selected via filters to include recent (last 5 years) and English- or Spanish-language studies in adults. This process yielded 18 results; the papers were downloaded and analyzed to meet additional inclusion criteria, such as having an observational design and having been performed with validated methods for urine culture, susceptibility testing, and gene expression.

We included papers that reported on fluoroquinolone resistance for uropathogens causing community-acquired urinary tract infections. A library was created in EndNote and duplicates were removed (2 duplicates were found). Data extracted from selected studies included the country of the study, study setting, time frame, patient demographics, uropathogens isolated, antimicrobial susceptibility, and mutation prevalence related to antimicrobial resistance. Some studies were eliminated or excluded because it was not possible to distinguish patterns of resistance between community-acquired and hospital-acquired UTIs, or because the resistance pattern included mixed pathogens.

## 3. Results

The number of papers selected at each stage of the review process is described in the PRISMA flow diagram (Figure 3). A total of 19 studies were included in this systematic review, which were selected from a wide distribution of studies around the world.

The geographical distribution of the studies’ origins was North America in five studies, South America in three studies, Europe in five studies, and Asia in six studies.

With respect to the length of time related to the observation period of the studies, we found a minimum period of 1 month, such as the case of Brazil [22] and Romania during June 2018 [23]. The longest period was 5 years, which was performed in Canada from 2015 to 2019 [24] and in Saudi Arabia during the same period of time [25].

Among the nineteen studies, only five mentioned phylotypes, with the most common phylotype reported being B2.

Ten studies referred to antimicrobial resistance genes (ARGs), and the most common AR genes were *blaCTX-M* and *blaTEM.* Only 11 studies mentioned ESBL-producing strains. Two studies from Asia (Turkey and Egypt) reported genes encoding plasmid-mediated quinolone resistance proteins (PMQRs), such as *qnr*, *aac (6′)-lb-cr*, *quepA*, and *oqxAB*. The most common PMQRs were *qnrS* and *qnrB.* The countries with the highest resistance rates to FQ were Bangladesh (69%), followed by Iran (55.6%).

More specific information related to countries, specific populations, time, and resistance rates to FQ, PMQRs, and ESBLs can be found in detail in Table 1 and in the Discussion Section.

## 4. Discussion

Fluoroquinolone resistance emerged in the mid-1990s because of their extensive use; for the following few decades, resistance to these antibiotics continued to increase [25]. Up-to-date information points toward a decreased incidence in the United States (7%), a higher incidence in Europe and Asia (28%), and the highest rates in developing countries [9]. Other results revealed a high rate of fluoroquinolone resistance in UPEC in India (>60%). In 2014, in Europe, resistance to fluoroquinolone was reported in 22% of strains. Specifically in Poland, resistance to fluoroquinolone was observed in approximately 30% of UPEC strains, and in Germany from 2013 to 2014, the percentage of ciprofloxacin-resistant UPEC strains was 17.3%. In Brazil from 2013 to 2014, 18.8% of outpatients were resistant to ciprofloxacin [39]. Our review is consistent with these previous findings, although the percentages have changed over the last five years. Nevertheless, the global burden of drug-resistant infections is enormous, both in terms of mortality and financial costs. Globally, 700,000 people are estimated to die each year due to drug resistance, and approximately 23,000 of these deaths occur in the United States. The Centers for Disease Control and Prevention estimate the cost of antimicrobial resistance in the United States to be USD 55 billion each year [40].

The presence of a drug-resistant strain in an individual may arise from direct infection with a drug-resistant strain, or alternatively the sensitive strain may mutate during a single infection and become resistant during transient treatment with fluoroquinolones [41].

The phenotypic antibiotic resistance of uropathogens causing UTIs is important for informing decision-making and determining the causes of such changes in incidence according to the region. Table 1 presents a summary of the studies found addressing this issue.

### 4.1. Reported Resistance Rates and Community-Acquired Uncomplicated Urinary Tract Infections

A recent systematic review related to *E. coli* resistance to fluoroquinolone in uCA-UTIs in women revealed that ciprofloxacin resistance in isolates varies by country but nevertheless tends to increase over time, especially in the United Kingdom, where it increased from 0.5% in 2008 to 15.3% in 2016; in Spain, it increased from 22.9 to 30.8%, and in Germany it increased from 8.5 to 15.1%. Furthermore, in Asia, a substantial increase in ciprofloxacin resistance from 25% in 2008 to 40% in 2014 was reported, with the highest percentage of resistance in Bangladesh (69%) [37]. Even more, when we analyzed reports of fluoroquinolone resistance after the EMA warning in 2018, we found two studies reporting significant resistance in Iran (55.6%) [20] and Romania (15%) [23].

A retrospective study conducted in China reported that the resistance rate to ciprofloxacin in UPEC ranged from 55 to 70% during the study period from 2012 to 2019 [42].

In North America, resistance increased from 4% in 2008 to 12% in 2017 [43]. Our review revealed a decrease in resistance from 18% to 14.2% in recurrent UTIs, and 8.6% in non-recurrent UTIs in the United States [28], whereas in Canada, resistance rates seem to be higher and related to specific regions [24]. The observed decreased prevalence in fluoroquinolone resistance might be associated with the FDA warning on fluoroquinolone use (July of 2016); although this is mostly speculative, it certainly is of interest for future studies in this country.

Fluoroquinolone resistance rates in patients with uCA-UTIs in Turkey increased from 20–30% in 1996 to 52% in adults with CA-UTIs in 2020 [2].

In our country, a previous study performed in 2020 reported that less than 40% of isolated strains were resistant to ciprofloxacin, although this resistance was detected in ESBL producers [29].

Additional studies performed in Latin America, specifically in Uruguay, reported that up to 13.6% of *E. coli* isolates causing CA-UTIs in adults were resistant to ciprofloxacin in 2014 [44]. However, in our review, we report on trends in fluoroquinolone resistance ranging from 15 to 29% for the treatment of community-acquired uUTIs [30,31].

### 4.2. Plasmidic Fluoroquinolone Resistance Genes and Phylotypes Prevalence

A study performed on samples from 168 Iranian women with community-acquired uncomplicated UTIs demonstrated a high prevalence of resistance to cefotaxime and aztreonam, and the lowest rates of resistance were associated with ciprofloxacin. In the same study, the phylotypes of the isolated *E. coli* strains were B2 > D > E > F > B1 > C > A [20].

In 2013, a study performed in Algeria reported the presence of mutations in *E. coli* associated with resistance to fluoroquinolones, and they reported that up to 56.66% of isolates carried mutations in *gyrA* and *parC* [45].

More recently, a study performed in Egypt (2016–2017) reported higher frequencies of the gene *qnrB* (100%); the *qnrS* gene was detected in 74.4% and the *qepA* gene was detected in 10% of samples from community- and hospital-acquired UTIs [38].

A study performed in Turkey aimed at investigating the frequency of plasmid-mediated fluoroquinolone resistance (PMFR) genes in ESBL-producing *E. coli* strains isolated from pediatric and adult patients with UTIs; the study revealed that up to one-third (92/258) of all the isolates carried at least one PMFR gene, with the most prevalent being *qnrS*, which was present in 67.4% of the isolates [2]. A second study from Turkey, performed in 2019, revealed that the most prevalent clone responsible for UTIs was *E. coli* clone O25b-ST131 (22%), which bore in up to 73% of cases the gene *CTX-M-1* and in 37% of cases the gene *CTX-M-15* [36].

Studies from Venezuela and Brazil detected the same genes, *blaTEM* and *blaSHV*, in *E. coli* isolates, whereas the *blaCTX-M* gene was found only in ESBL-producing isolates, mainly in community-acquired UTIs [22,30].

A meta-analysis performed in 2022 reported an increase in the incidence of phylotype B2 during the years 2014–2020 worldwide. The authors’ discussion focused on the presence of additional risk factors, such as host species, nutrition, type of infection, and geographical region, to explain the variation in the phylotype frequency of *E. coli* [46]. With respect to this phylotype, a study in Bucharest, Romania, performed in 2019, revealed that up to 35% of the causative agents for CA-UTIs was phylotype B2, 27% was B1, and 22% was phylotype A. The authors also investigated ESBL genes and reported *blaCTM-M* in 42.25%, *blaTEM* in 38.02%, and *blaSHV* in 19.71% [23]. In Iran, during the spring of 2021, the prevalence of *E. coli* resistance genes was determined, with the following percentages of the resistance genes: *blaTEM*, 89.6%; *blaCTX-M*, 44.3%; *blaSHV*, 6.6%; and *blaCMY*, 0.9%. The most frequent phylotype was B2 in 29.2% and D in 17.9% [20].

### 4.3. Risk Factors Identified for Resistance

Fluoroquinolone-resistant infections are often associated with known risk factors such as recent use of fluoroquinolones, living in long-term care facilities or hospitalized patients, pregnancy, being male, being immunocompromised or having current or pre-existing functional or anatomical urinary tract abnormalities, advanced age, large prostate size, high postvoid residual volume, hypertension, and diabetes [29,44,47].

A systematic review found that in women with uCA-UTIs caused by isolates resistant to fluoroquinolones, risk factors could be postmenopausal status and UTI recurrence [43]. Being female could be a risk factor due to the anatomy of the female tract. The female urethra, which is shorter and closer to the anus, is the main reason why UTIs are more common in adolescents, adults, and older women [3].

After the present review was conducted, we observed a relatively high percentage of resistance to fluoroquinolones in Asian countries. These countries might have common risk factors that make them vulnerable to a higher and rapid increase in resistance rates, such as frequent nonprescription antimicrobial use [2]. A study that explored the consequences of antibiotic prescribing regulations in different regions in Canada reported that regions with better fluoroquinolone susceptibility were areas with more strict antibiotic prescribing regulations [24].

Additional studies performed in Europe have suggested regional differences with respect to ESBL-positive isolates, with the identification of variables related to subjects (age), the environment, and antibiotic use in the last 3 months to determine the presence of these variants [48].

With respect to environmental factors, others have explored the possibility of regions with circulation of already resistant genotypes, which might be present in contaminated food and promote person-to-person propagation of the resistant pathogen [22].

For older males, who are susceptible to recurrent UTIs, another risk factor related to bladder outlet obstruction is prostate enlargement; additionally, recurrent UTIs and previous antibiotic use have both been associated with antibiotic resistance [37]. Furthermore, travel and swimming have been identified as risk factors for the acquisition of ESBL-producing *E. coli* strains [20].

The present review provides an overview of the different aspects of antimicrobial resistance, including gene and phylotype prevalence, the mechanism of resistance, rates of CA uUTI by country, and risk factors for the development of FQ resistance. This information could help in the development of new strategies for antibiotic prescription, especially FQ. While the implementation of measures to address and decrease the FQ resistance rates has taken place, alternative therapeutic options should be considered, as recommended in current guidelines for nitrofurantoin, trimethoprim–sulfamethoxazole, fosfomycin, and fluoroquinolones for uUTIs in men, after reviewing local patterns of resistance.

Furthermore, stewardship programs promote coordinated measures aimed at optimizing the use of diagnostic techniques, favoring the adoption of adequate therapeutic, clinical, and preventive decisions. Stewardship incorporation in microbiological diagnosis on a routine basis can improve patient outcomes while strengthening the role of clinical microbiologists in the management of infectious diseases [49].

## 5. Conclusions

Fluoroquinolone resistance rates highlight the importance of local resistance monitoring and investigations of underlying mechanisms, including local prescription regulations, which are used in the setting of complicated infections, restrict the use of prescriptions in veterinary medicine, and highlight the importance of the rational use of antimicrobials. This review offers some information related to resistance genes and bacterial phylotypes that can be useful for comparing trends in the coming years.

## Figures and Tables

**Figure 1 microorganisms-12-02320-f001:**
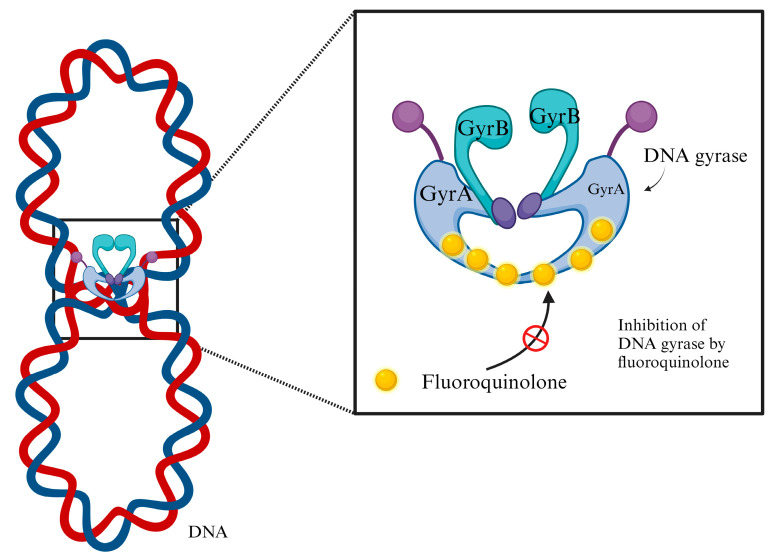
Mechanism of action of fluoroquinolones. *E. coli* DNA gyrase (which acts by unwinding the DNA double helix) is composed of two A subunits and two B subunits encoded by *GyrA* and *GyrB*, respectively. The A subunits carrying the “codon trimming” functions of gyrase are the site of action of fluoroquinolones. The drug inhibits (red circle) gyrase-mediated DNA supercoiling at concentrations that are clearly related to those required to inhibit bacterial proliferation. Created in BioRender.com.

**Figure 2 microorganisms-12-02320-f002:**
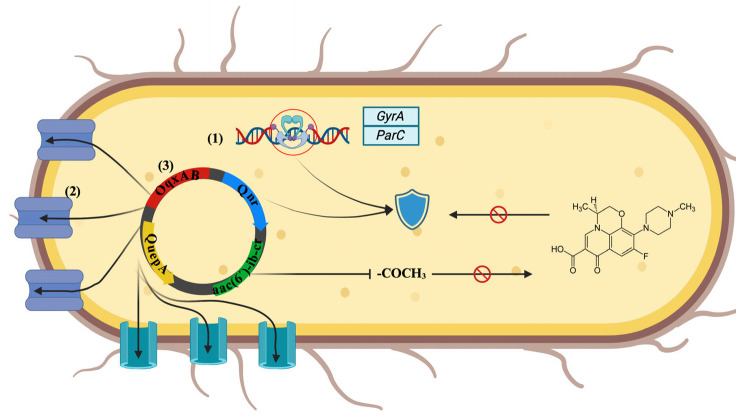
Mechanisms of fluoroquinolone resistance in *E. coli*. (**1**) Chromosomal mutations induce changes in drug targets, such as genes that encode *GyrA*, and *ParC*; (**2**) mutations related to reducing the drug concentration in the bacterial cytoplasm through the overexpression of efflux pumps and downregulation of porins; and (**3**) the expression of genes that encode plasmid-mediated quinolone resistance proteins (PMQR), such as *qnr* (protects DNA gyrase and topoisomerase IV), *aac(6’)-ib-cr* (acetylates the fluoroquinolone so that it is inactivated), *quepA*, and *oqxAB* (overexpressing multidrug efflux pumps or decreasing the permeability of outer membrane proteins). Created in BioRender.com.

**Figure 3 microorganisms-12-02320-f003:**
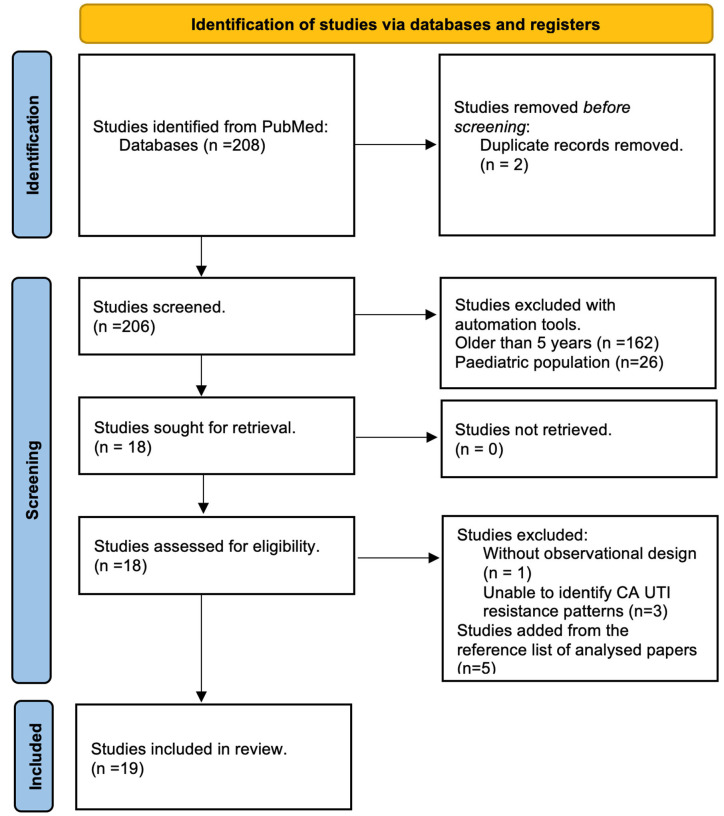
PRISMA flow diagram for systematic reviews.

**Table 1 microorganisms-12-02320-t001:** Summary of the studies addressing fluoroquinolone resistance patterns in patients with urinary tract infections.

Country/Year	Population	Period	Isolates	Resistance to FQ	Phylotype	AR Gene	ESBL
Canada 2020 [26]	11,333 isolates from community-acquired UTIs (507 nosocomial)	April 2010–December 2014	*E. coli*	CIP: 18.4%	NA	NA	NA
Canada 2022 [24]	591 antibiograms from patients with CA UTIs	2015–2019	*E. coli*	FQ: 11–63%	NA	NA	NA
USA 2019 [27]	73 cases of CA-UTIs	April 2015 to May 2016	*E. coli* (59%)	CIP: 18%	NA	NA	NA
USA 2024 [28]	68,033 cases of non-recurrent uUTIs and 12,234 cases of recurrent UTIs	October 2015–February 2020	* E. coli *	Recurrent UTIs FQ: 14.2% Non-recurrent uUTIs FQ: 8.6%	NA	NA	Recurrent UTIs 5.9% and non-recurrent UTIs 4.1%
Mexico 2020 [29]	296 inpatients with community-acquired UTIs.	2018–2019	* E. coli *	CIP: 30%	NA	NA	25.6% of which 89% were resistant to CIP
Venezuela 2019 [30]	43 isolates from patients with uUTIs and 60 complicated UTIs	January–June 2014	*E. coli*	CIP: 29.7%	NA	*blaTEM* (65.4%) *blaCTX-M* (34.6%) *blaSHV* (23.1%)	20.4%
Brazil 2020 [22]	499 isolates from patients with CA-UTIs	November 2015	*E. coli*	CIP: 20%	B2 (30%) D (23%) A (13%) F (12%) B1 (9%) C (8%) E (3%)	*blaTEM* (10%) *blaSHV* (3%) *blaCTX-M* (93%)	8%
Argentina 2021 [31]	1740 cases of CA-UTIs	January 2016 to December 2017	*E. coli* (80%)	CIP: 15.2%	B2 D	*blaCTX-M*	0.2% of which 56.9% were resistant to CIP
United Kingdom 2020 [32]	836 *E. coli* isolates for resistance	September 2017–August 2018	*E. coli*	CIP: 50.7%	B2 D	*blaCTX-M* *pAmpC* *blaCMY* *blaDHA*	NA
France 2020 [33]	190 women with non-severe community-onset pyelonephritis.	March–August 2018, and April–August 2019	*E. coli* (84%)	FQ: 3–17%	NA	NA	NA
Germany 2023 [34]	386 isolates from female patients with uUTIs	January 2017–December 2019	*E. coli*	FQ: 5.2% CIP: 8.2%	NA	NA	NA
Poland 2020 [35]	796 isolates from 332 patients with recurrent lower UTIs	2016–2018	*E. coli* (40%)	CIP: 39.9%	NA	NA	9%
Romania 2019 [23]	787 patients with CA-UTIs	June 2018	*E. coli* (91%)	LEV: 14.86% CIP: 14.99%	B2 (35%) B1 (27%) D (16%) A (22%).	*blaCTX-M* (42.3%) *blaTEM* (38.0%) *blaSHV* (19.7%) *fimH* (93.9%) *hlyD* (44.3%) *afaBC* (38.2%) *hlyA* (12.4%) *cnf-1* (7.7%).	9%
Turkey 2019 [36]	101 from CA UTIs and nosocomial UTIs	April–August 2018	*E. coli*	CIP: 50.98%	NA	*blaCTX-M-1* (73%) *blaCTX-M-15* (37%) *O25b-ST131* (22%)	50.49%
Turkey 2020 [2]	141 adult outpatients with UTIs	1 June 2015 to 31 March 2016	*E. coli*	FQ: 35.92%	NA	*qnrS* (67.4%) *aac (6′)-1b-cr* (42.4%) *qnrB* (7.6%)	51.78%, of which 29.12% were resistant to FQ.
Bangladesh 2022 [37]	4500 patients with community-acquired UTIs	September 2016–November 2018	*E. coli* (51.6%)	CIP: 69%	NA	NA	NA
Egypt 2019 [38]	440 isolates from patients with UTIs	July 2016–March 2017	*E. coli* (64%)	CIP: 19.2% NOR: 19.2% OFX: 19.2%	NA	*qnrB* (62.9%) *qnrS* (46.9%) *qepA* (6.3%)	NA
Iran 2023 [20]	168 CA uncomplicated UTIs	Spring season of 2021	*E. coli*	CIP: 55.6%	B2 (29%) D 17.9% E 14.1% F 9.4% C 6.6%	*blaTEM* (89.6%) *blaCTX-M* (44.3%) *blaSHV* (6.6%) *blaCMY* (0.9%)	52.8%
Saudi Arabia 2021 [3]	428 patients with a positive urine culture (≥10^5^ CFU/mL).	January 2015– December 2019	* E. coli* (45%)	CIP: 53.1%	NA	*bla_CTX-M*	23.7%, of which 47.8% were resistant to CIP

FQ: fluoroquinolone. CIP: ciprofloxacin. LEV: levofloxacin. NOR: norfloxacin. OFX: ofloxacin. NA: not applicable. ESBL: extended-spectrum beta-lactamase.

## Data Availability

No new data were created or analyzed in this study.

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
