# Peer review of "Fluoroquinolone Resistance in Escherichia coli Causing Community-Acquired Urinary Tract Infections: A Systematic Review"

_microorganisms, 2024, doi:10.3390/microorganisms12112320_

Round 1

Reviewer 1 Report (Previous Reviewer 2)

Comments and Suggestions for Authors

The manuscript "Fluoroquinolone resistance in Escherichia coli causing community-acquired urinary tract infections: a systematic review" presents a systematic review on the prevalence of fluoroquinolone resistance in E. coli causing community-acquired urinary tract infections (CA-UTIs). The review aims to address the knowledge gap regarding the prevalence of E. coli phylotypes and genes responsible for fluoroquinolone resistance in CA-UTIs. The authors conducted a systematic search in PubMed and included 19 studies that met their inclusion criteria. The review highlights the global distribution of fluoroquinolone resistance, with the highest rates observed in developing countries. It also discusses the various mechanisms of fluoroquinolone resistance, including chromosomal mutations, efflux pump overexpression, and plasmid-mediated quinolone resistance genes. The review further explores the prevalence of different E. coli phylotypes and resistance genes in CA-UTIs, with B2 being the most common phylotype and blaCTX-M and blaTEM being the most prevalent resistance genes. The authors also identify several risk factors associated with fluoroquinolone resistance, such as recent fluoroquinolone use, long-term care facility residence, hospitalization, pregnancy, male sex, immunocompromised status, urinary tract abnormalities, advanced age, large prostate size, high post-void residual volume, hypertension, and diabetes. The review concludes by emphasizing the importance of local resistance monitoring, investigating underlying resistance mechanisms, implementing antibiotic prescription regulations,and promoting the rational use of antimicrobials. The authors also suggest considering alternative therapeutic options, such as Nitrofurantoin, Trimethoprim-Sulfamethoxazole, and Fosfomycin, for the treatment of uncomplicated UTIs. The review provides valuable insights into the prevalence and mechanisms of fluoroquinolone resistance in E. coli causing CA-UTIs. The information presented in this review can aid in developing effective strategies for antibiotic prescription and managing the growing problem of fluoroquinolone resistance. The authors have conducted a comprehensive search and analysis of relevant studies, and the review is well-organized and clearly written. The inclusion of tables and figures further enhances the clarity and understanding of the presented data. The review also highlights the need for further research to explore the regional variations in resistance rates and identify additional risk factors associated with fluoroquinolone resistance. The incorporation of more recent investigations (DOI 10.3390/life1401010), would enhance the comprehensiveness of this review by providing a more contemporary perspective on the evolving landscape of fluoroquinolone resistance. The inclusion of the latest research findings would ensure that the review remains current and accurately reflects the most recent trends in this critical area of antimicrobial resistance. Overall, this systematic review is a valuable contribution to the field of infectious diseases and provides important information for clinicians, researchers, and public health professionals involved in the management and prevention of UTIs.

Author Response

Thank you for your kind comments, and taking the time to read and review our manuscript.  

We are interested in reading and adding updated information as you mentioned, however the DOI provided (DOI 10.3390/life1401010) does not retrieve any article. Can you please provide more information so that we can find it and add it to our manuscript?

Thank you.

Reviewer 2 Report (New Reviewer)

Comments and Suggestions for Authors

The manuscript entitled: ‘Fluoroquinolone resistance in Escherichia coli causing community-acquired urinary tract infections: a systematic review’ reviews quinolone resistance in UTI.

 The authors review several studies published in recent years, and conclude that local monitoring and regulation of local prescriptions is important.

In this regard, the authors do not take into account that both the American FDA and the European Medicines Agency (EMA) in 2016 and 2018, respectively, warned against the use of quinolones in non-serious infections, including UTI, and recommended restricting their use.  This situation should be taken into  account and included in the revision

FDA updates warnings for fluoroquinolone antibiotics | FDA (https://www.fda.gov/news-events/press-announcements/fda-updates-warnings-fluoroquinolone-antibiotics)

Summary of the EMA public hearing on quinolone and fluoroquinolone antibiotics (europa.eu)

https://www.ema.europa.eu/en/documents/report/summary-ema-public-hearing-quinolone-and-fluoroquinolone-antibiotics_en.pdf

In addition, the studies selected by the authors, for the most part, study periods close to the alerts issued by the agencies. It would have been interesting to use more recent study periods to compare the situation before and after the alert.

 On the other hand, in the introduction, it seems that the most important target of resistance is GyrA, whereas the two enzymes GyrA and ParC are important for generating high rates of resistance.

Other important issue is that the flow  diagram representing the included studies, it is not clear and the numbers do not add up, and in the final number of included studies there are 19 when 18 have been selected.

Author Response

The authors review several studies published in recent years, and conclude that local monitoring and regulation of local prescriptions is important. In this regard, the authors do not take into account that both the American FDA and the European Medicines Agency (EMA) in 2016 and 2018, respectively, warned against the use of quinolones in non-serious infections, including UTI, and recommended restricting their use.  This situation should be taken into account and included in the revision

FDA updates warnings for fluoroquinolone antibiotics | FDA (https://www.fda.gov/news-events/press-announcements/fda-updates-warnings-fluoroquinolone-antibiotics)

Summary of the EMA public hearing on quinolone and fluoroquinolone antibiotics (europa.eu)

https://www.ema.europa.eu/en/documents/report/summary-ema-public-hearing-quinolone-and-fluoroquinolone-antibiotics_en.pdf

Response: Thank you for your comments. A paragraph was added to inform on these warnings in the introduction section, both websites were added to the reference list.

In addition, the studies selected by the authors, for the most part, study periods close to the alerts issued by the agencies. It would have been interesting to use more recent study periods to compare the situation before and after the alert.

Response: We reviewed the selected studies to identify those that were reporting FQ resistance after the warnings and find only 2 for the European Warning and 1 (questionable date) related to the FDA warning. We added two paragraphs to reflect on this in the discussion section, subsection 4.1 Reported resistance rates and comm unity-acquired uncomplicated urinary tract infections.

On the other hand, in the introduction, it seems that the most important target of resistance is GyrA, whereas the two enzymes GyrA and ParC are important for generating high rates of resistance.

Response: Yes, we agree that both mutations are important factors for resistance to FQ, we have changed figure 2 to include both. They were both already mentioned in the text.

Other important issue is that the flow diagram representing the included studies, it is not clear and the numbers do not add up, and in the final number of included studies there are 19 when 18 have been selected.

Response: We are sorry, the flow chart was added as an image but some of the information was removed for some reason, we pasted the image in a different format, and now you can see the complete process of identification of studies, numbers add up, 5 studies were added from the reference list of analyzed papers.

Reviewer 3 Report (New Reviewer)

Comments and Suggestions for Authors

The Review entitled "Fluoroquinolone resistance in Escherichia coli causing community-acquired urinary tract infections: a systematic review" performed by Ruiz-Lievano et al., is a well conducted review and a good manuscript that is relevant to the field. 

This revised version has been improved from the previous but there are still some minor issues listed below:

- L85, L94, the genes must be written in lower case shape (GyrA -> gyrA, ParC -> parC)

- L107, Please consider changing "insertion of sequence" to "Insertion sequence"

-L132, please shift "Mesh" to "MESH"

- Table 1, please revise the entire table text, the bacterial names must be written in italics, but the specie must be written in lower case (E. Coli -> E. coli)

- Table 1, there is an overlaped data of the blaTEM gene percentage in the Venezuelan line

- L209, please change "two studies with significant resistance" to "two studies reporting significant resistance"

- L238, shift QnrS to qnrS

Author Response

The Review entitled "Fluoroquinolone resistance in Escherichia coli causing community-acquired urinary tract infections: a systematic review" performed by Ruiz-Lievano et al., is a well conducted review and a good manuscript that is relevant to the field. 

This revised version has been improved from the previous but there are still some minor issues listed below:

L85, L94, the genes must be written in lower case shape (GyrA -> gyrA, ParC -> parC)

Reponose: We have made this change.

L107, Please consider changing "insertion of sequence" to "Insertion sequence"

Response: Thank you for this observation, we have made the change.

L132, please shift "Mesh" to "MESH"

Response: Thank you, we have made the change. 

Table 1, please revise the entire table text, the bacterial names must be written in italics, but the specie must be written in lower case (E. Coli -> E. coli)

Response: Table 1, has been edited.

Table 1, there is an overlaped data of the blaTEM gene percentage in the Venezuelan line

Response: Table 1 has been edited, according to your suggestions. 

L209, please change "two studies with significant resistance" to "two studies reporting significant resistance"

Response: We made this change. 

L238, shift QnrS to qnrS

Response: we made the adjustment, thank you. 

Round 2

Reviewer 2 Report (New Reviewer)

Comments and Suggestions for Authors

see attachment

Author Response

The authors have addressed all the questions. 

However there are one question that is difficult to understand for me. 

In the final part of the flow chart, the authors had 18 studies assessed for eligibility, and discarded 9, and finally included 19?. 

Response: the last part of the figure explains the from the 18 studies assessed for eligibility 4 were discarded and 5 were added from the reference list of analyses papers. So 18-4= 14, and then 14+5=19. That is how we finally included in the review a total of 19 studies.

This manuscript is a resubmission of an earlier submission. The following is a list of the peer review reports and author responses from that submission.

Round 1

Reviewer 1 Report

Comments and Suggestions for Authors

This is a systematic review on the development of resistance of E. coli to fluoroquinolones in the outpatient setting. The topic is interesting, and the article contains carefully crafted, good figures.

However, I noticed some points that should be improved:

·         English revision required

·         Title: The authors should consider changing the title to focus on E. coli, as the main part of the article addresses E. coli resistance.

·         Introduction:

o   The distinction between uncomplicated and complicated urinary tract infections, as well as infections in men and women, pregnant women, etc., should be specified in more detail.

o   Comment from my side: “The European Centre for Disease Prevention (ECDC) has identified the misuse of ABs as one of the leading causes of AMR. According to the ECDC, misuse occurs mainly in the following three scenarios: (i) unnecessary prescription of ABs for viral infections, (ii) prescription of broad-spectrum ABs as a result of ignorance of the causative pathogen, and (iii) incorrect doses, frequencies, or treatment durations of AB administration.” (Deininger et al. 2022)

o   The precise indication for fluoroquinolones according to guidelines (e.g., EAU) should be described.

·         Methodology:

o   Was a PROSPERO registration carried out? If not, why not?

·         Results:

o   The Results section should be revised; the first sentences are hardly comprehensible.

o   The PRISMA flowchart is somewhat disorganized, not well-ordered, and needs revision.

o   Table 1: Abbreviations should be mentioned in the table caption.

o   The results are presented only in table format; they should also be explained.

Comments on the Quality of English Language

Editing required. 

Author Response

We would like to thank the time invested by reviewer 1, as it has improved the quality of our manuscript. Please read specific responses to each comment. 

English revision required

Response: Thank you. The English revision was done by a native English speaker and corrections were made throughout the manuscript.

  • Title: The authors should consider changing the title to focus on E. coli, as the main part of the article addresses E. coli resistance.

Response: The title has been changed, thank you for this suggestion.

                        Introduction:

  • The distinction between uncomplicated and complicated urinary tract infections, as well as infections in men and women, pregnant women, etc., should be specified in more detail.

Response: Thank you for your recommendation, this information has been added to the introduction section. References 5-7 were added as well.  

Comment from my side: “The European Centre for Disease Prevention (ECDC) has identified the misuse of ABs as one of the leading causes of AMR. According to the ECDC, misuse occurs mainly in the following three scenarios: (i) unnecessary prescription of ABs for viral infections, (ii) prescription of broad-spectrum ABs as a result of ignorance of the causative pathogen, and (iii) incorrect doses, frequencies, or treatment durations of AB administration.” (Deininger et al. 2022)

Response: Thank you, the comment was added to the introduction section, as well as the additional reference.

  • The precise indication for fluoroquinolones according to guidelines (e.g., EAU) should be described.

Response: We added information published in both: guidelines of the Infectious Diseases Society of America (IDSA) and the European Centre for Disease Prevention (ECDC) in the introduction section.

                         Methodology:

  • Was a PROSPERO registration carried out? If not, why not?

Response: We did not design a protocol before data extraction, and therefore did not register the protocol in PROSPERO database. As the data extraction has already taken place, currently is not recommended to perform the registry. However, we will take this into consideration for future systemic reviews conducted by our group.

  • Results:
  • The Results section should be revised; the first sentences are hardly comprehensible.

Response: We rephrased most paragraphs from the results section.

  • The PRISMA flowchart is somewhat disorganized, not well-ordered, and needs revision.

Response: We apologize, something went wrong during submission and the format changed. We converted to text to an image, to prevent future changes in the format.

  • Table 1: Abbreviations should be mentioned in the table caption.

Response: Abbreviations were included below the table 1.

  • The results are presented only in table format; they should also be explained.

Response: The table is part of the discussion section, were we have more detailed explanation of results obtained from this review. Results were broadly described to avoid repetitive information.

Reviewer 2 Report

Comments and Suggestions for Authors

The article provides a comprehensive overview of fluoroquinolone resistance in uropathogenic E. coli causing community-acquired urinary tract infections. The review is well-structured, beginning with an introduction that establishes the significance of UTIs and the concern of rising antibiotic resistance. The authors clearly define CA-UTI and outline the common treatment options, emphasizing the advantages of fluoroquinolones while acknowledging the growing resistance problem. 

The mechanisms of fluoroquinolone resistance are explained in detail, covering both chromosomal mutations and plasmid-mediated mechanisms. The role of mobile genetic elements in the spread of resistance genes is also discussed, along with the classification of E. coli strains into phylotypes. The authors briefly touch upon the potential contribution of antibiotic use in veterinary medicine to resistance development.

The review methodology is sound, following PRISMA guidelines and utilizing a systematic search strategy in PubMed. The inclusion and exclusion criteria are well-defined, ensuring the relevance of the selected studies. The results are presented clearly, summarizing the findings of 19 studies from various regions. The discussion section effectively contextualizes the results, comparing them with previous findings and highlighting the global trends in fluoroquinolone resistance. The authors address the clinical implications of resistance and emphasize the need for local resistance monitoring and rational antibiotic use.

The authors should provide the registration information for this systematic review, including the register name and registration number, as this is a standard practice for systematic reviews to ensure transparency and reduce bias

The review could be strengthened by expanding the discussion on the implications of the findings for clinical practice and public health. For instance, the authors could elaborate on the potential impact of fluoroquinolone resistance on treatment guidelines and the need for alternative therapeutic options. Additionally, discussing the role of antibiotic stewardship programs in curbing resistance would be valuable.

Overall, this review is a valuable contribution to the literature on fluoroquinolone resistance in UPEC causing CA-UTIs.It provides a thorough overview of the current knowledge and identifies areas for future research. The clarity of the writing and the well-organized structure make it accessible to a broad audience, including clinicians, researchers, and public health professionals.

Comments on the Quality of English Language

Minor editing of English language required.

Author Response

We would like to thank the time invested by reviewer 2, as it has provide with some reassuring messages, while at the same time, improved the quality of our manuscript. Please read specific responses to each comment. 

The authors should provide the registration information for this systematic review, including the register name and registration number, as this is a standard practice for systematic reviews to ensure transparency and reduce bias

Response: We did not design a protocol before data extraction, and therefore did not register the protocol in PROSPERO database. As the data extraction has already taken place, currently is not recommended to perform the registry. Will take this into consideration for future systemic reviews of our group.

The review could be strengthened by expanding the discussion on the implications of the findings for clinical practice and public health. For instance, the authors could elaborate on the potential impact of fluoroquinolone resistance on treatment guidelines and the need for alternative therapeutic options. Additionally, discussing the role of antibiotic stewardship programs in curbing resistance would be valuable.

Response: Thank you for this observation, we obtained more information related to these topics and included in the discussion section. We believe this addition has enriched our discussion and might provide a broaden view of the problem as well as insights into a broader sense for a solution.

Overall, this review is a valuable contribution to the literature on fluoroquinolone resistance in UPEC causing CA-UTIs.It provides a thorough overview of the current knowledge and identifies areas for future research. The clarity of the writing and the well-organized structure make it accessible to a broad audience, including clinicians, researchers, and public health professionals.

Response: Thank you very much for your comments.